



# Combining short-range dispersion simulations with fine-scale meteorological ensembles: probabilistic indicators and evaluation during a [85]Kr field campaign

Youness El-Ouartassy[1,2], Irène Korsakissok[2], Matthieu Plu[1], Olivier Connan[3], Laurent Descamps[1], and Laure Raynaud[1]

[1]CNRM, University of Toulouse, Météo-France, CNRS, 31057, Toulouse, France.
[2]Institut de Radioprotection et de Sûreté Nucléaire (IRSN), PSE-SANTE/SESUC/BMCA, F-92260, Fontenay-aux-Roses, France.
[3]Institut de Radioprotection et de Sûreté Nucléaire (IRSN), PSE-ENV/SRTE/LRC, F-50130, Cherbourg-En-Cotentin, France.

**Correspondence:** Youness El-Ouartassy (Youness.el-ouartassy@meteo.fr)

**Abstract.** Numerical models of atmospheric dispersion are used for predicting the health and environmental consequences of nuclear accidents, in order to anticipate the countermeasures necessary to protect the populations. However, the simulations of these models suffer from significant uncertainties, arising in particular from input data: weather conditions and source term. To characterize weather uncertainties, it is essential to combine a well-known source term data and meteorological ensembles to generate ensemble dispersion simulations which has the potential to produce different possible scenarios of radionuclides dispersion during emergency situations. In this study, the fine-scale operational weather ensemble AROME-EPS from Météo-France is coupled to the Gaussian puff model pX developed by French Institute for Radiation Protection and Nuclear Safety (IRSN). The source term data is provided by Orano La Hague reprocessing plant (RP) that regularly discharges [85]Kr during the spent nuclear fuel reprocessing process. Then, to evaluate the dispersion results, a continuous measurement campaign of [85]Kr air concentration was recently conducted by the Laboratory of Radioecology in Cherbourg (LRC) of IRSN, around RP in the North-Cotentin peninsula.

This paper presents a probabilistic approach to study the meteorological uncertainties in dispersion simulations at local and medium distances (2-20 km). As first step, the quality of AROME-EPS forecasts is confirmed by comparison with observations from both Météo-France and IRSN. The following step is to assess the probabilistic performance of the dispersion ensemble simulation, as well as the sensitivity of dispersion results to the method used to calculate atmospheric stability fields and their associated dispersion Gaussian standard deviations. Two probabilistic scores are used: Relative Operating Characteristic (ROC) curves and Peirce Skill Score (PSS).

The results show that the stability diagnostics of Pasquill provides better dispersion simulations. In addition, the ensemble dispersion performs better than deterministic one, and the optimum decision threshold (PSS maximum) is 3 members. These results highlight the added value of ensemble forecasts compared to a single deterministic one, and their potential interest in the decision process during crisis situations.



## 1 Introduction

Accidental releases of radioactive pollutants into the atmosphere can have serious impact on human health and environment (Aliyu et al., 2015; Nie et al., 2021). The dispersion of radionuclides released into the atmosphere depends on the physico-chemical properties of the released substances, the emission parameters (e.g. source elevation, timing and duration of the release) and meteorological conditions at the accident site (e.g. wind speed and direction) (Girard et al., 2014). In order to forecast the dispersion of radionuclides in the early phase of nuclear accidents and to support decisions and warnings, atmospheric dispersion models (ADM) are commonly used to predict the transport of radioactive pollutants through the atmosphere as well as the quantities of radioactive material deposited on the ground (Korsakissok et al., 2013). This information is essential for decision makers in order to anticipate the countermeasures necessary to protect the population against contamination.

### 1.1 Uncertainties and ensemble simulations

The outputs from ADM simulations suffer from significant uncertainties that limit the confidence in them when they are used in an operational context (Korsakissok et al., 2020; Leadbetter et al., 2020). The three main sources of these uncertainties have been discussed by Rao (2005) and by Mallet and Sportisse (2008). The first one is related to the source term, which is an essential input data. For prognosis of potential releases, it may be defined from a priori assumptions (pre-defined source term), modelling of physical processes at stake within the reactor, along with knowledge of the damaged installation status. In case of an ongoing or past release, when observations are available in the environment, the source term can be reconstructed by inverse methods. For this purpose, IRSN (French Institute for Radiation Protection and Nuclear Safety) has developed inverse modelling methods, which are mathematics-based methods aiming to minimize the difference between ADM outputs and in-situ measurements (Saunier et al., 2013, 2020).

The second main source of uncertainty is related to the meteorological forecasts that are given as input to ADM. Weather information used for dispersion prediction is, frequently, provided by numerical weather predictions (NWP) as 3-D or 4-D physical fields. To take into account the meteorological uncertainties on dispersion simulations, two methods have commonly been used. The first one is by adding random perturbations to weather inputs (Girard et al., 2014, 2020). The second one is by using meteorological ensembles (Straume et al., 1998).

Some studies have used operational ensemble prediction systems (EPS) as input for dispersion models in the case of the Fukushima accident (Sørensen et al., 2016; Kajino et al., 2019; Le et al., 2021), and others for hypothetical nuclear accident scenarios (Sørensen et al., 2019; Korsakissok et al., 2020; Leadbetter et al., 2021). These studies include either meteorological uncertainties only, or sometimes both meteorological and source term uncertainties. All these studies were carried out at long distance and the ensembles used to represent weather uncertainties had coarse spatial and temporal resolution. For example, De Meutter et al. (2016) studied the use of meteorological ensembles at hemispheric scale to predict radioxenon peaks coming from radiopharmaceutical facilities, and De Meutter and Delcloo (2022) at continental scale for the same application, using ECMWF ERA5 ensemble at a horizontal resolution of 63 km. Le et al. (2021) investigated the dispersion of radionuclides at Japan scale in the case of the Fukushima disaster, using the operational ECMWF-ENS (Leutbecher and Lang, 2014) with a



spatial resolution of approximately 25 × 25 km and 3 h time steps, using several source terms from literature. In Le et al. (2021)
      and De Meutter and Delcloo (2022), an evaluation of the dispersion ensembles was performed by comparison to radiological
      observations in the environment. In a hypothetical case study, Leadbetter et al. (2021) explored the uncertainties coming only
      from weather conditions at synoptic scale, by using the operational Met Office's EPS named MOGREPS-G (Tennant and
      Beare, 2014) with a spatial resolution of approximately 20 × 20 km and 3-h time steps. Although this approach allowed to

examine the ability of meteorological ensembles to perform atmospheric dispersion results more skillful than results produced
      with deterministic meteorology, it did not evaluate the performance of the ensemble dispersion simulations in the case of a
      realistic release. While most applications of meteorological ensembles cited above were focused on hemispheric or continental
      scale, the impact of meteorological uncertainty on dispersion forecasts at local scale (in the range of 2 to 20 km) has received
      less attention. With the development of kilometer-scale EPS (Bouttier et al., 2012), the feasibility and interest for such studies

are rising. High-resolution meteorological ensembles were used in the case of a fictious nuclear release (Sørensen et al., 2017;
      Korsakissok et al., 2020), but no comparison to observations was made. The realistic performance of ADM outputs can be
      assessed only by using well-known real source term combined with reliable tracer measurements appropriate for the studied
      scale, as discussed in Section 1.2.

      The third source of uncertainty arises from approximations for resolving atmospheric processes in the ADM, such as, for

instance, turbulent diffusion and deposition (Leadbetter et al., 2015; Girard et al., 2016). A possible approach to include
      these model-related uncertainties is to use a multi-model ensemble. This approach consists in using a set of different ADM
      to construct an ensemble of simulations, either with identical or different input data, to represent the modelling uncertainties
      (Galmarini et al., 2004b, a). It has been used for various applications, including the Fukushima accident (Draxler et al., 2015;
      Sato et al., 2018). In this paper, we use a single ADM, but the influence of model-related variables such as atmospheric turbulent

parameters is discussed.

### 1.2 Dispersion datasets at local scale

There is a large panel of atmospheric dispersion tracer experiments for model validation at local scale (Olesen, 1998), both
for rural and urban areas. However, most of the experiments were conducted within a few kilometers of the source. There is a
lack of tracer measurement experiments studies ranging from the short to medium distances (2 km-20 km). At such scale, the

$^{85}$Kr can be a good tracer since it is an inert gas with a long half-life ($\tau_{1/2}$=10.7 years) and its radioactive decay negligible at
      these distances. The main sources of the $^{85}$Kr in the atmosphere are reprocessing plants of spent nuclear fuel, from which the
      $^{85}$Kr release can be known with a reasonable accuracy. As an example, the work conducted in the Laboratory of Radioecology
      in Cherbourg (LRC) of IRSN, presented by Connan et al. (2013), is one of the rare studies that explored the dispersion of
      radionuclides at distances between 5 km and 50 km. In this previous paper, continuous $^{85}$Kr measurements at 1 minute time

period were carried out at three stations and combined with well-known discharge data which were provided by the nuclear
      fuel reprocessing plant of Orano La Hague (later called RP), located in the North-Cotentin peninsula (north-western France).
      These data were then used to perform dispersion simulations, but, with only three stations, the dataset was not large enough
      to capture the spatial spread of released radioactive material. From these previous studies arose the need for a campaign using





more observation stations, spatially representative of the area, along with a longer time-period to make the conclusions more
statistically robust. These previous studies also showed that the assumption of homogeneous meteorological data, using a
single meteorological observation as input, was responsible for a large part of simulation errors (Korsakissok et al., 2016), thus
highlighting the need to account for meteorological uncertainties.

### 1.3  Objectives of the paper

The main purpose of the present article is to investigate the impact of the meteorological uncertainties on local-scale dispersion.
The operational high-resolution meteorological ensembles AROME-EPS (Bouttier et al., 2012) and AROME deterministic
NWP (Seity et al., 2011) of Météo-France are used as input of the IRSN short-range Gaussian puff model pX (Soulhac and
Didier, 2008; Mathieu et al., 2012; Korsakissok et al., 2013) around RP facility at local scales (less than 20 km). In this area,
there is a dense weather observation network (from both IRSN and Météo-France) that has been used to validate AROME-EPS
ensembles before combining them with the dispersion model. Measurements of $^{85}$Kr air concentration at eight fixed points
located at various distances, from 2 km to 20 km, and at various orientations from RP facility were carried out by IRSN in
the framework of the DISKRYNOC project (DISpersion of KRYpton in the NOrth-Cotentin). This dataset is presented for the
first time in this paper, and is used to evaluate the probabilistic performance of ensemble dispersion simulations. Thus, the
originality of this work can be summarized in three points: (i) the use of a unique and original dataset of continuous data of
$^{85}$Kr air concentration measurements (every 1 minute or 10 minutes) over a relatively long period, (ii) the evaluation of an
ensemble of dispersion simulations using a fine-scale meteorological ensemble with in-situ observations, (iii) an innovative
method developed to assess the probabilistic performance of the dispersion ensembles.

The outline of the article is as follows: in Section 2, the source term, the observations and the models used in the study are
described. Section 3 presents the verification of AROME-EPS against wind measurements, and then in Section 4 the ensemble
dispersion simulations are presented and discussed. Conclusion and perspectives are provided in Section 5.

## 2  Case study, data and models

### 2.1  Case study

The present study focuses on the dispersion of $^{85}$Kr at short and medium distances (less than 20 km), in the North-Cotentin
peninsula located in the North-West of France territory (Fig. 1). This geographical area is where the nuclear fuel reprocessing
plant of Orano La Hague (later called RP) is located (Fig. 1). $^{85}$Kr is a $\beta^-$ and $\gamma$ emitting radioactive noble gas that is
naturally present in the environment, but mainly released into the atmosphere during the reprocessing of spent nuclear fuel.
The potential interest of the La Hague area is that the release rate of $^{85}$Kr emitted by the RP into the atmosphere is known with
a good accuracy. In addition, there is a sufficient density of meteorological measurements combined with $^{85}$Kr radiological air
concentration measurements. Meteorological measurements are carried out by Météo-France on a regular basis. IRSN's LRC
laboratory regularly performs meteorological and radiological measurements in the framework of measurement campaigns.





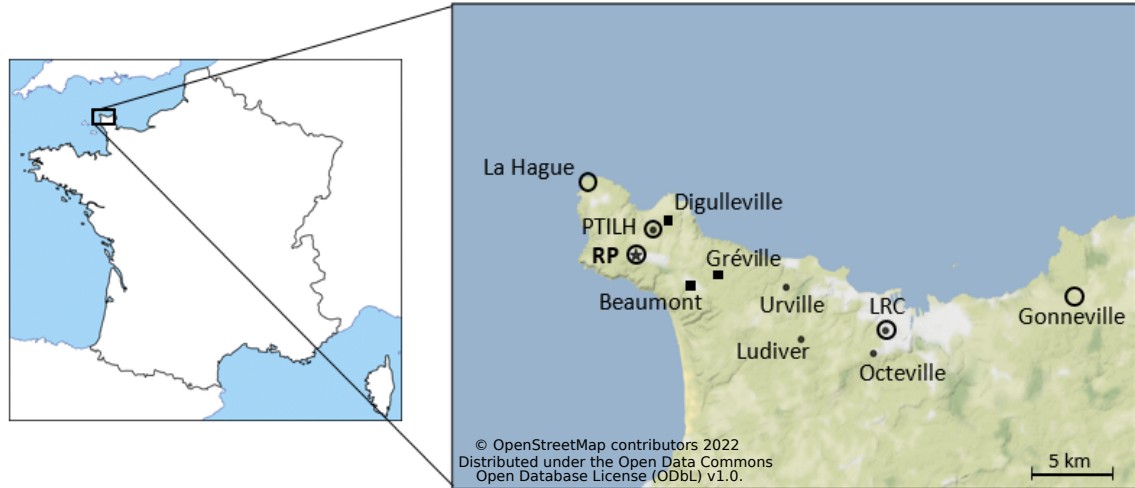

**Figure 1.** Location of North-Cotentin peninsula (left panel) and map of the monitoring sites (right panel). The dots and squares indicate the locations of the $^{85}$Kr measurement stations carried out by IRSN and RP, respectively, as part of the DISKRYNOC campaign. The RP facility location is marked with a star. The circles indicate the locations of the 3D-wind measurement sites (from IRSN or Météo-France).

Additional meteorological and air concentration measurements are carried out by Orano for the environmental monitoring of the RP. For these reasons, the La Hague experimental site is an ideal environment for the study and validation of atmospheric dispersion simulations.

Past validation studies conducted in this framework have shown that dispersion simulation results are quite sensitive to the meteorological data used as input (Connan et al., 2013). The North-Cotentin peninsula of La Hague is a rocky area of approx-

imately 15 km long and 5 km wide, surrounded by the sea (Fig. 1). Such a complex terrain leads to spatially heterogeneous wind fields that may be difficult to accurately forecast. Therefore, this case study should provide good insights to examine the influence of meteorological uncertainties on atmospheric dispersion simulations.

## 2.2  Source term of $^{85}$Kr

The La Hague RP has two production units called UP2-800 (1.87941° W, 49.67705° N) and UP3 (1.87606° W, 49.67705° N).

Each of the two units has a stack for the discharge of $^{85}$Kr with a height of 100 m and the two stacks are 200 m apart (Leroy et al., 2010).

During the reprocessing process of spent nuclear fuel, $^{85}$Kr is intermittently released into the atmosphere from stacks of the plant for periods of 30 to 45 min, separated by approximately 10 minutes periods without releases. Depending on the industrial activity, long periods (a few hours to a few days or weeks) without releases are frequent. Both plants can operate separately

and the release can come from one or both plants. $^{85}$Kr release fluxes (measured at a frequency of 10 min) were provided by RP for the study period. The sum of the amounts of $^{85}$Kr released from UP2 and UP3 units, over regular 10 minutes intervals





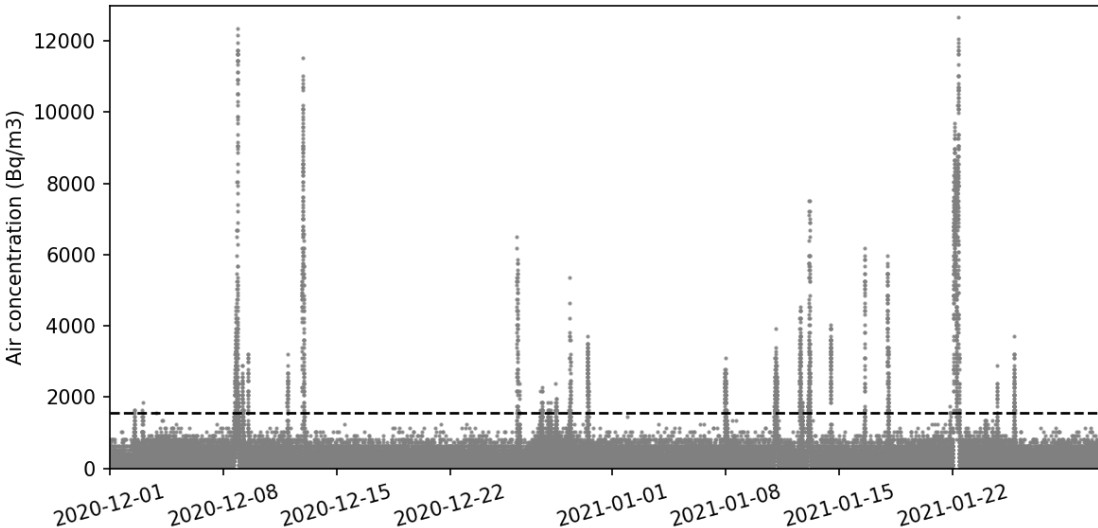

**Figure 2.** $^{85}$Kr air concentration measurements carried out at LRC (Figure 1) from 01 Dec. 2020 to 31 Jan. 2021. The horizontal dashed line shows the air concentration threshold (1545 $Bq.m^{-3}$) above which peaks are considered.

constitutes an accurate and reliable source term. The radioactive concentration of the released $^{85}$Kr depends on the burn-up of the reprocessed spent fuel and the processing rate of the plant (Connan et al., 2013).

In this paper, the atmospheric dispersion of $^{85}$Kr is studied along the continuous two-month period ranging from 01 Decem-
ber 2020 to 31 January 2021. This period comprises the detection of an important number of $^{85}$Kr events at all measurement sites (Fig. 2) due to favourable wind direction.

## 2.3 Measurements campaign of $^{85}$Kr in the North-Cotentin

The IRSN routinely monitors $^{85}$Kr air concentrations close to the RP to study the transfer of radionuclides in the environment, but also to validate the atmospheric dispersion models and improve the understanding of radionuclides dispersion in various
atmospheric conditions (Maro et al., 2002, 2007; Leroy et al., 2010; Connan et al., 2014). $^{85}$Kr is a very good tracer of atmospheric dispersion in short and medium distances since it is an inert gas (noble gas), which means it does not generate chemical or physical reactions, so it does not get depleted by rain (wet scavenging) or by dry deposition processes. In addition, $^{85}$Kr has a sufficiently long half-life ($\tau_{1/2}$=10.7 years) for its radioactive decay to be negligible at short and medium distances.

Since November 2020, IRSN has been carrying out a continuous $^{85}$Kr air measurement campaign in several locations chosen
at different distances and directions from RP, as part of the DISKRYNOC project. This project aims to provide a comprehensive new observational dataset for model validation purposes. In this study, this dataset is used to acquire feedback on the use of meteorological ensembles to quantify the associated uncertainties. The 8 closest air sampling locations from RP used in this article, described in Table 1 and shown in Fig. 1, are : PTILH, Urville, Ludiver, Octeville, LRC, Digulleville, Beaumont and



| Stations | Lon/Lat | Height above ground | Distance from RP facility | Measurements time step |
|---|---|---|---|---|
| PTILH | 1.8733° W, 49.6949° N | 1.5 m | 2 km | 1 min |
| Urville | 1.7431° W, 49.6607° N | 1.5 m | 10.4 km | 1 min |
| Ludiver | 1.7283° W,49.6297° N | 1.5 m | 12.7 km | 1 min |
| Octeville | 1.6579° W, 49.6211° N | 1.5 m | 17.7 km | 1 min |
| LRC | 1.6458° W, 49.6347° N | 2 m | 18 km | 1 min |
| Digulleville | 1.8595° W, 49.7001° N | 2 m | 2.6 km | 10 min |
| Beaumont | 1.8358° W, 49.6613° N | 2 m | 4.2 km | 10 min |
| Gréville | 1.8097° , 49.6682° N | 2 m | 5.2 km | 10 min |

**Table 1.** Description of the 8 localisations of $^{85}$Kr air concentration measurement stations used in this article.

Gréville. The IRSN is the owner of the first five stations, while the last three are Orano stations. The measurements have
been carried out since mid-November 2020 and are expected to extend over approximately 18 months. This extended time
period should provide a significant number of observations at all measurement sites. It should compensate for periods without
reprocessing activity or with a wind direction towards the sea. Typical values of $^{85}$Kr air concentrations in these stations
range from tens to thousands $Bq.m^{-3}$ (depending on the distance from RP, wind direction, plant reprocessing activities and
atmospheric conditions). Continuous measurements are being performed every minute in the IRSN stations and every 10
minutes in the RP stations.

The activity concentration in the air is determined by $\beta$ counting in a Berthold LB123 or LB134 gas proportional counter
calibrated with a common $^{85}$Kr source specially fabricated (Gurriaran et al., 2001). This method is only useful in near fields
(less than 30 km) where the $^{85}$Kr air concentration is sufficiently high. The same method has been used by Connan et al. (2014)
and it has been documented by Gurriaran et al. (2004).

## 2.4 3-D wind observations

Evidence from past studies has shown that 3-D wind field is one of the most sensitive meteorological parameter for atmospheric
dispersion models (Girard et al., 2014). For this reason, the performance of the AROME-EPS forecasts in terms of wind speed
and direction should be assessed before they may be used for atmospheric dispersion. For this purpose, four kinds of observation
data of wind have been used:

- The real-time ground observation acquisition network of Météo-France called RADOME, which has about 550 automatic
  ground observation stations spread over the whole French territory, among which two stations are located in the study
  area and are shown in Fig. 1: La Hague (1.9398° W, 49.7251° N) located ∼2.5 km from the RP plant, and Gonneville
  (1.4635° W, 49.6526° N) located ∼31 km from the RP plant. These stations provide continuous hourly measurement
  data of 10 m wind, temperature, humidity, rainfall and surface solar radiation fields.



– Vertical wind profile measured by Doppler Lidars (LIght Detection And Ranging). Atmospheric Lidars are currently
        used for atmospheric measurements of aerosols and wind, and thus allow for climate monitoring, air quality or cloud
        monitoring (Werner, 2005; Wu et al., 2022). A Doppler Lidar (version Leosphere Windcube 2) was recently installed by
        the IRSN's LRC laboratory, on the PTILH (Instrumented Technical Platform of La Hague) measurement site, located
        ∼2 km from the RP plant (Table 1). This Lidar provides wind data (speed and direction) at 10 minutes intervals on 13
vertical levels: 40, 60, 80, 100, 120, 140, 160, 180, 200, 220, 240, 260 and 280 m.

        – Ultrasonic measurements acquired by Sodar (Sonic Detection And Ranging), which is a remote sensing instrument often
        used in meteorology for the 3-D acquisition of wind fields (speed and direction) on several vertical levels, using Doppler
        effect on sound waves levels (Tamura et al., 2001).

        The Sodar measurements used in this work come from the instrument located ∼200 m West (1.8901° W, 49.6800° N) of
the RP facility, which provides measurements on six vertical levels: 0, 10, 50, 100, 150 and 200 m.

        – Ultrasonic measurements by LRC's anemometer (1.6458° W, 49.6347° N) installed at a height of 13 m above the ground.
        This instrument provide 10 minutes wind measurements.

Thus, five wind measurement points are available and used to evaluate the AROME-EPS meteorological ensemble over the
two-month period of this work. This validation process has been done near the surface and on several vertical levels of the
atmospheric boundary layer (ABL), as the knowledge of the evolution of the meteorological fields through out the lower
atmosphere with a good accuracy is beneficial for a short distance atmospheric dispersion model to describe the physical
processes (e.g. turbulence) occurring in the ABL.

## 2.5    Description of AROME and AROME-EPS

The Météo-France NWP model AROME used in this study is summarized in Table 2 and extensively documented in Seity
et al. (2011). AROME is a non-hydrostatic kilometer scale NWP limited area model. This model covers a geographical domain
of about 1000 × 1000 km centred over the French territory with 90 vertical levels and horizontal resolution of 1.3 km. The
lateral and upper boundary conditions are provided by the operational global NWP model ARPEGE (Courtier et al., 1991)
of Météo-France. AROME runs four times per day up to at least 42 h range, starting from the initial times 0000, 0600, 1200
and 1800 UTC. Its 3D-VAR data assimilation scheme (Brousseau et al., 2011) is a state-of-the art assimilation algorithm, that
produce analyses at 1.3 km resolution by correcting the model state at hourly time steps using different kinds of meteorological
observations (in-situ ground-based measurements, radar reflectivities and winds, satellite radiances, among others.).

   The AROME-EPS (Bouttier et al., 2016) used in this article is a 16 members ensemble based on the AROME model at 2.5 km
(Table 2). The ensemble runs four time per day, up to at least 45 h range, at 0300, 0900, 1500 and 2100 UTC. The 16 AROME-
EPS perturbed initial conditions are built from the AROME 3D-var analyses on which perturbations from the Ensemble Data
Assimilation (EDA) at 3.25 km resolution are added (Raynaud and Bouttier, 2016). The AROME EDA comprises 25 members
that are obtained by perturbing the observations and the model state during the assimilation process. The outputs from both
are combined and interpolated to 2.5 km to produce the initial conditions of the AROME-EPS. The lateral and upper boundary



|  | AROME | AROME-EPS |
|---|---|---|
| Domain | Western Europe, centred on France (∼1000 x 1000 km) | |
| Size | Deterministic (1 forecast) | 16 members |
| Horizontal resolution | 1.3 km | 2.5 km |
| Vertical levels | 90 [5 m-10 hPa] | 90 [5 m-10 hPa] |
| Forecast initial time | 0000, 0600, 1200, 2100 UTC | 0300, 0900, 1500, 2100 UTC |
| Forecast range | 48 h, 42 h, 48 h, 42 h | 45 h, 51 h, 45 h, 51 h |

**Table 2.** Description of AROME and AROME-EPS.

conditions (Bouttier and Raynaud, 2018) are provided by the Météo-France ARPEGE-EPS operational global EPS (Descamps et al., 2015).

Besides the initial errors, forecast uncertainty also arises from the dynamic part of the model (e.g. spatial and temporal discretization of the equations that represent phenomena whose characteristic scale is larger than the mesh size), or from the physical part of the model (e.g. corrective terms added to the dynamic equations to take into account the effect of phenomena whose scale is smaller than the mesh size). To account for model uncertainties in the AROME-EPS forecasts, the Stochastically Perturbed Parametrization Tendencies (SPPT) scheme is used (Palmer et al., 2009; Bouttier et al., 2012). This method consists in adding random perturbations to the model physics tendencies.

### 2.6 Description of the pX model

The IRSN's Gaussian puff model pX used in this work is part of the operational platform C3X (Tombette et al., 2014), which is used by IRSN Emergency Response Center in case of an accidental radioactive release. pX is used to simulate the atmospheric dispersion of radionuclides on short and medium distances [500 m-50 km] (Korsakissok et al., 2013; Mathieu et al., 2012). The principle of such a dispersion model is based on the following assumptions:

- The release comes from a point source,

- A continuous release can be discretized into a series of puffs transporting a given amount of pollutants,

- Within each puff, the meteorological variables can be considered homogeneous,

- The concentration of pollutant in the puff can be represented by a Gaussian law in each of the three directions (Appendix B).

For an instantaneous release of a mass $Q$ of a given radionuclide, the concentration $c$ at a given point $(x, y, z)$ and a time $t$ is given by:

$$c(x, y, z, t) = \frac{Q}{(2\pi)^{2/3} \sigma_x \sigma_y \sigma_z} \exp\left[-\frac{1}{2}\left(\frac{(x-x_0)^2}{\sigma_x^2} + \frac{(y-y_0)^2}{\sigma_y^2} + \frac{(z-z_0)^2}{\sigma_z^2}\right)\right] \tag{1}$$





Where $\sigma_x$, $\sigma_y$ and $\sigma_z$ are the Gaussian standard deviations of the diffusion of the puffs over time in the three directions of space and $(x_0, y_0, z_0)$ is the position of the mass center of the puff, which is transported by the mean wind flow. If $x$ is the mean wind direction, and $\boldsymbol{U}$ is the mean wind speed in the $x$ direction, then the position of the centre of the puff at each time $t + dt$ from its position at time $t$ is :

$$\boldsymbol{x_0}(t + \Delta t) = \boldsymbol{x_0}(t) + \boldsymbol{U}(t)\Delta t \tag{2}$$

This advection scheme allows to transport the puffs' mass centers through a non-stationary and heterogeneous wind field. In addition, the puffs are growing over time, to represent the plume's mixing by atmospheric turbulence. This is represented in Eq. (1) by the standard deviation $\sigma_x$, $\sigma_y$ and $\sigma_z$ that increase over time. This increase of plume spread depends on the atmospheric stability, and is described by empirical standard deviation laws. In the pX model, the laws of Doury (Doury, 1976) or Pasquill (Pasquill, 1961) can be used. In this work, Pasquill stability was determined using two methods : (i) Pasquill-Turner (Turner, 1969) and (ii) the temperature gradient between 10 m and 100 m in the meteorological forecasts (Seinfeld and Pandis, 1998). That is, three stability diagnoses which are compared in this work (Appendix A).

For a continuous emission of release rate $q_s$ (in $Bq.s^{-1}$) that is discretized into a series of $N$ puffs, each puff $i$ containing a mass $Q_i = q_s\Delta t$, the concentration $c$ at a given point is computed as the sum of the contribution of all puffs:

$$c(x, y, z, t) = \sum_{i=1}^{N} c_i(x, y, z, t) \tag{3}$$

where $c_i$ is given by Eq. (1).

Finally, the mass of the material transported by the puff is depleted by wet and dry deposition as well as by radioactive decay. In our case, the transported mass will be assumed to remain constant over time, since $^{85}$Kr is an inert gas (no deposition) who has a long half-life (no radioactive decay at short-distance) as shown previously. Equation (1) is also modified to take into account reflections on the ground and ABL height under certain conditions. Specifically, reflections on the ABL height are considered in unstable situations (when a capping inversion is assumed).

## 3 AROME-EPS verification

### 3.1 Scores for AROME-EPS verification

Before coupling the numerical weather predictions from AROME-EPS to the pX model, it is necessary to evaluate them in order to have an exhaustive overview of their quality and to take it into account in the interpretation of atmospheric dispersion simulations. For this purpose, comparative evaluation scores based on the observations of 3D-wind speed and direction described in Section 2.4 have been calculated:

**Bias:** In order to identify the systematic deviations of AROME-EPS meteorological ensemble forecasts from the observations, the bias over all days of the period of interest for a variable $X$ is calculated at each forecast range $t$, by the following





equation:

$$Bias(t) = \frac{1}{N_{day}} \sum_{d=1}^{N_{day}} \left( \langle X_{mod}^{(t,d)} \rangle - X_{obs}^{(t,d)} \right) \tag{4}$$

Where $N_{day}$ is the number of days of the interest period. $\langle X_{mod}^{(t,d)} \rangle$ is the AROME-EPS ensemble mean at forecast range $t$ on day $d$ and $X_{obs}^{(t,d)}$ is the observed value at the same instant.

**Spread-skill:** As shown by Fortin et al. (2014), the ability of an ensemble to represent simulation errors can be evaluated by comparing, at each forecast range, the skill (or Root Mean Square Error, RMSE) of the ensemble mean and its spread (Spd), the latter calculated relative to the ensemble mean (Raynaud et al., 2012; Charrois et al., 2016). For a variable $X$,

the ensemble Spd and RMSE terms are defined, at each forecast range $t$, as follows:

$$Spd(t) = \sqrt{\frac{1}{N_{day}} \sum_{d=1}^{N_{day}} \frac{1}{N_{ens}-1} \sum_{n=1}^{N_{ens}} \left( X_{mod,n}^{(t,d)} - \langle X_{mod}^{(t,d)} \rangle \right)^2} \tag{5}$$

$$RMSE(t) = \sqrt{\frac{1}{N_{day}} \sum_{d=1}^{N_{day}} \left( \langle X_{mod}^{(t,d)} \rangle - X_{obs}^{(t,d)} \right)^2} \tag{6}$$

Where $N_{ens}$ represents the ensemble size ($N_{ens}$=16 in the case of AROME-EPS). The value of variable $X$ given by the ensemble member $n$ at the forecast range $t$ is $X_{mod,n}^{(t,d)}$. This diagnostic can be summarized by calculating the spread–skill

ratio, which should be as close to 1 as possible. Values less than 1 (respectively greater than 1) indicate that the ensemble is underdispersive (respectively overdispersive).

### 3.2 Model-to-data comparison of AROME-EPS

The evaluation of the quality of AROME-EPS predictions was carried out over the two-months period considered in this study (Dec. 2020-Jan. 2021). This evaluation process is done independently for each of the 0300, 0900, 1500 and 2100 UTC

forecasts for all stations described in Section 2.4. The results for all forecasts and stations are similar. Therefore, only the results of the 1500 UTC forecast are shown here for two configurations: (i) at 10 m height (two RADOME stations: La Hague and Gonneville) and (ii) at several levels of ABL for stations where wind vertical profile measurements are available (Lidar at PTILH and Sodar at the RP site).

Figure 3 shows the ensemble biases in terms of 10 m wind speed and direction aggregated from La Hague and Gonneville

stations. In the case of wind speed, the ensemble mean is above the observation for most of the forecast ranges, resulting in a slight systematic bias which varies between $-0.2$ and $1.75$ m.s$^{-1}$. Both forecasts and observations (and eventually the resulting bias) evolution shows a marked diurnal cycle. The maximum bias is around 1500 UTC, corresponding to forecast range 25 h, with approximately no bias in the first forecast range. For the wind direction there is a good average performance of the model as shown by the good agreement between the averages of the forecasts and observations. The mean bias of the model oscillates

around zero, with maximum and minimum values of $+10°$ and $-15°$, respectively.




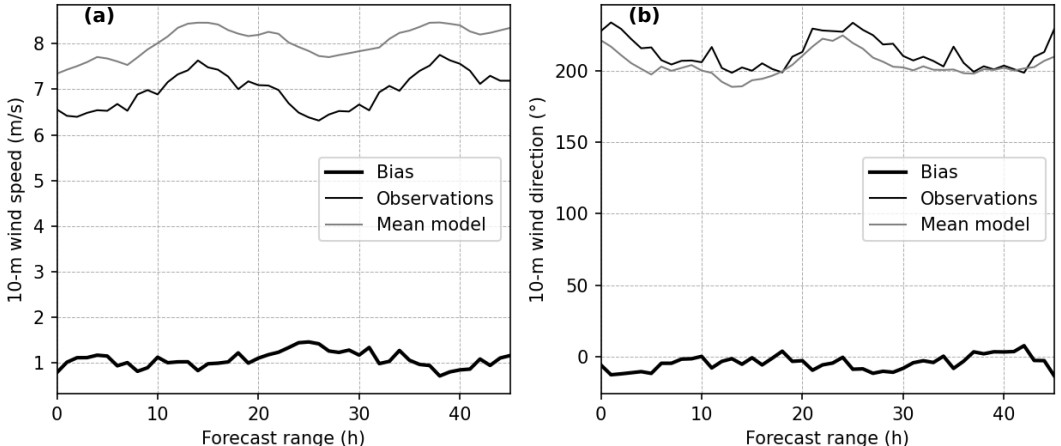

**Figure 3.** Ensemble mean, observations and the resulting bias for both 10 m wind speed (a) and direction (b) aggregated from the two ground measurement stations La Hague and Gonneville, as a function of forecast range.

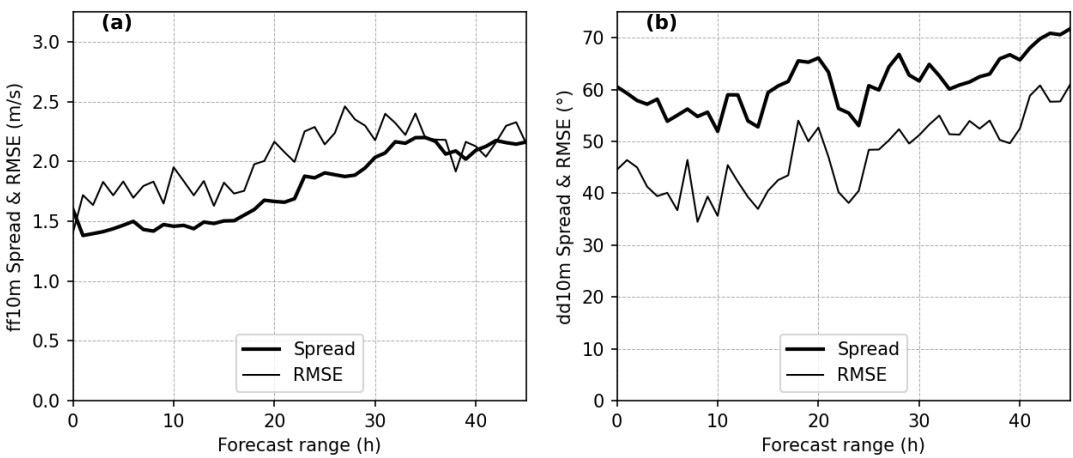

**Figure 4.** Ensemble spread and RMSE of the ensemble mean forecast for both 10 m wind speed (a) and direction (b) aggregated from the two ground measurement stations La Hague and Gonneville, as a function of forecast range.

Figure 4 shows the ensemble spread and skill evolution over forecast range. For wind speed, the spread of the ensemble is consistent with the RMSE with respect to the observations, with a slight underdispersion, while for wind direction the ensemble spread is above the RMSE at all forecast ranges. This overdispersion in terms of wind direction should be kept in mind when interpreting the ensemble atmospheric dispersion simulations.





| Levels (m) | Wind speed and direction bias (m.s$^{-1}$) | Wind speed and direction spread-skill ratio |
|---|---|---|
| 40 | 0.948, 1.696 | 0.959, 1.416 |
| 60 | 0.606, 0.852 | 1.034, 1.410 |
| 80 | 0.448, 0.913 | 0.062, 1.412 |
| 100 | 0.364, 1.002 | 1.089, 1.416 |
| 120 | 0.217, 0.581 | 1.116, 1.418 |
| 140 | 0.141, 0.490 | 1.134, 1.412 |
| 160 | 0.095, -0.421 | 1.145, 1.407 |
| 180 | 0.053, -0.468 | 1.151, 1.402 |
| 200 | 0.062, -0.305 | 1.171, 1.386 |
| 220 | 0.042, -0.915 | 1.182, 1.356 |
| 240 | 0.023, 0.247 | 1.200, 1.393 |
| 260 | -0.021, -0.790 | 1.216, 1.395 |
| 280 | -0.078, -1.246 | 1.226, 1.444 |

**Table 3.** The AROME-EPS ensemble bias and spread-skill ratio averaged over forecast range in the vertical levels observed by Lidar located in the PTILH site.

To complement this evaluation at 10 m height, it is worth examining the quality of the AROME-EPS meteorological ensembles at different vertical levels in the lower atmosphere. To do so, the bias and the spread-skill ratio have been calculated at several vertical levels above ground. The results at the PTILH station are presented in Fig. 5 and summarized in Table 3. The wind speed forecasts are slightly less biased and overdispersive at higher altitudes than at lower ones, with an overdispersion more pronounced in the earlier forecast ranges. The latter is probably due to an imperfect accounting of modeling and/or initial conditions uncertainties in the perturbation process. However, the bias at 40 m (Lidar measurements) and 10 m (in-situ measurements) are consistent, which means that probably the high bias in the lower layers is not due to Lidar measurement errors. It may be due to the representation of surface processes in AROME in this area which is characterised by a complexe orography and heterogeneous surfaces (sea and land). For the wind direction there is no significant dependency of biases and spread of the ensembles with respect to the altitudes.

To summarise, the assessment of the consistency of AROME-EPS forecasts showed that they perform well by comparison to wind speed and direction measurements in the North-Cotentin area for the selected period, despite slight errors in the wind speed forecast. The same conclusion was reached for deterministic AROME forecasts (not shown here), by calculating the bias from wind observations.




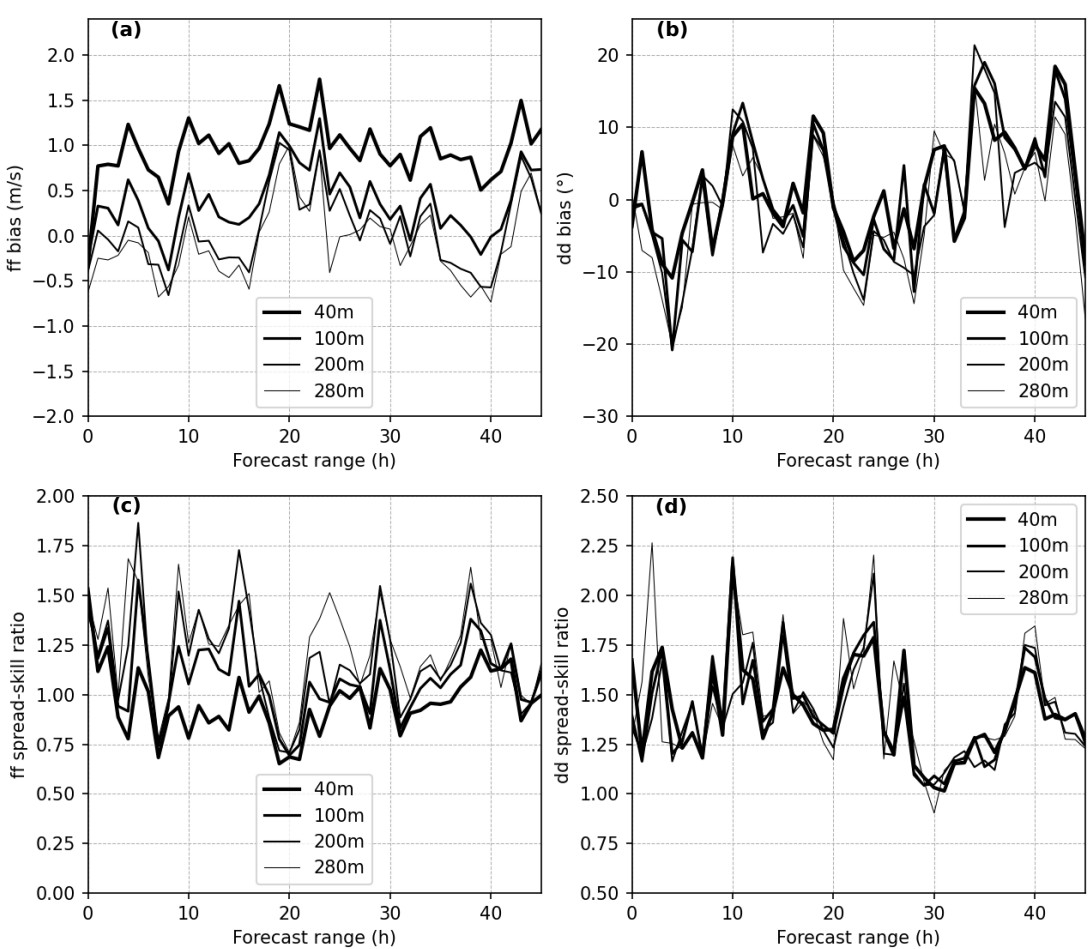

**Figure 5.** Bias (a, b) and spread-skill ratio (c, d) of the ensemble between 40 m and 280 m above ground, for both wind speed (a, c) and direction (b, d) measured by Lidar at the PTILH station, as a function of forecast range.

## 4 Analysis of the ensemble dispersion simulations

### 4.1 Coupling AROME-EPS and pX model

Once meteorological forecasts from AROME-EPS have been qualified as shown in section 3, they are coupled to the Gaussian dispersion model pX (section 2.6). This process consists in running in parallel several simulations with the pX model, each



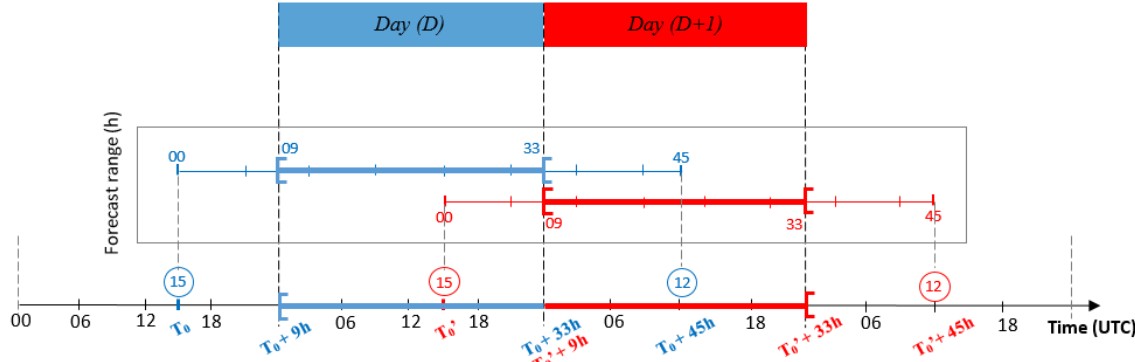

**Figure 6.** Illustration of meteorological forecasts used from AROME-EPS (in bold) as input to pX: the forecast starting from 1500 UTC on a day $D$ is used to cover the next day $D+1$.

using a different member of the AROME-EPS ensemble as input, along with the source term data provided by RP La Hague. This allows to generate an ensemble of dispersion simulations composed of 16 members (hereafter called the pX ensemble). Furthermore, in order to quantify the benefit of using ensembles instead of deterministic simulations, an additional pX simulation was performed using the deterministic weather forecast from AROME as input of the model. This simulation is called deterministic pX in the following.

### 4.1.1 Temporal continuity of AROME-EPS members

In the case of accidental releases that span a long time period, it is important to properly build continuous time series from several forecasts made at different initial times, without causing brutal jumps in spread between forecasts from two consecutive days. Besides, fine-scale weather forecasts are, usually, not available until few hours after their start. To deal with this issue, this study proposes to use the closest available forecast at the beginning of a day. In other words, to simulate a release occurring during a day $D$, the AROME-EPS ensemble forecasts starting from 1500 UTC of the day before $(D-1)$ were used. The first 8 forecast hours were skipped in order to let the perturbation grow, and the next 24 ones [09-32h] were used to cover the entire day. In the same way, the forecast starting from 1500 UTC of day $D$ was used to cover the day $D+1$. Figure 6 illustrates this cycle. Then, these intervals were combined to cover the three simulation periods detailed in Table 5, by connecting each member $i$ of day $D$ with member $i$ of day $D+1$.

Note that to perform pX deterministic simulations, the deterministic weather forecasts from AROME are built in the same way, by using the forecast starting from 1200 UTC. In this case the first 11 forecast hours were skipped and the next 24 h were used [12-35h].



### 4.1.2 pX simulation set-up

The calculation domain of pX is defined by the grid of meteorological forecasts, and the puffs that leave this domain no longer participate to the concentration calculations. Moreover, the concentration calculated on a point located on the border of the meteorological domain will only account for the contribution of the puffs inside. It is, therefore, necessary to define

a simulation domain whose borders are sufficiently far from the calculation points (i.e. [85]Kr measurements sites in Fig. 1). Thus, a $60 \times 60$ km domain centred on the source (i.e. RP) was defined, where the meteorological forecasts were interpolated on a Cartesian grid with 2.5 km of horizontal resolution, leading to a horizontal mesh of $24 \times 24$ cells. This process was accomplished by a weather pre-processor that was developed as part of this study.

Considering the objectives of this work, only the first 25 vertical levels outputs from AROME [10 m-3000 m] are used here

to cover the entire ABL. The ABL height is diagnosed from AROME forecasts as the lowest altitude where the turbulent kinetic energy is below $0.01$ m$^2$.s$^{-2}$. As a result, it may happen that this diagnostic reaches unrealistically low values, as low as 10 m. In order to avoid such low values and to ensure that the source emission does not occur above the ABL, a minimum value of 200 m is imposed to the ABL height before being applied to the pX simulations.

Even though the NWP forecasts are given with an hourly frequency, the pX simulations were performed in this study with

a time step of 10 minutes in order to better capture the temporal variations of the plume. Sensitivity tests showed that the pX simulations with the two Pasquill-based stability diagnoses (Pasquill-Turner and temperature gradient) gave very similar results with a very slight better performance of the diagnosis of temperature gradient. In the following, simulations with Pasquill (called pX-Pasquill) standard deviations will thus be computed with the latter stability diagnosis and will be compared with simulations using Doury standard deviations (called pX-Doury).

Finally, the effects of the complex topography (coastline, rocky terrain) and buildings on the plume dispersion may lead to downwash effects that are not explicitly taken into account by the Gaussian puff model. To compensate for this limitation, an effective height that differs from the physical stack height may be used as input. In this case, five values of effective height have been tested: 20, 50, 100, 150 and 200 m, and the most optimum simulations were obtained by using the physical stack height of 100 m.

### 4.2 Qualitative results and discussion


In this section, we illustrate the behavior of the dispersion simulations at two stations located at short and medium distances from the source. For this purpose, a short time period where the occurrence of few marked events was selected.

#### 4.2.1 Comparison of ensemble and deterministic dispersion results

Figure 7 and Figure 8 show an example of dispersion results at PTILH (2 km from the source) and LRC stations (18 km from

the source), respectively. There are observed peaks that the deterministic simulation does not reproduce while some ensemble members simulate them with acceptable accuracy. This highlights the potential interest of the ensemble approach compared to the deterministic one. In addition, visual analysis of the results shows that the pX-Pasquill simulations correctly predict most





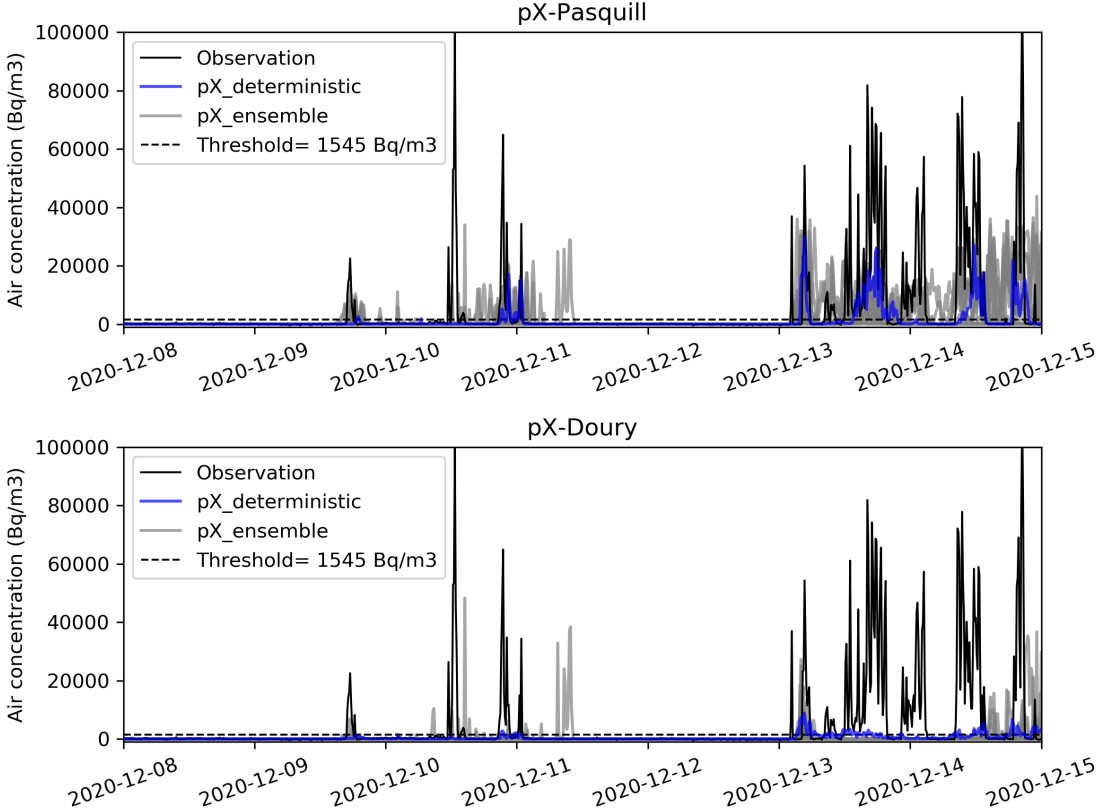

**Figure 7.** pX-Pasquill and pX-Doury ensemble and deterministic simulations of $^{85}$Kr air concentration compared to the observation in the first aggregated period (Table 5), at the PTILH station. The horizontal dashed line shows the air concentration threshold (1545 $Bq.m^{-3}$) above which peaks are considered.

peaks at PTILH station (Fig. 7), although with a tendency to underestimate their maximum value. At the same station, pX-Doury simulations show a strong underestimation of concentrations, resulting in a failure to forecast most observed peaks. As 360 the PTILH station is located only 2 km from the source, the release conditions (initial buoyancy and building downwash effects) largely influence the concentrations at this short distance. In our simulations, the use of the stack height (100 m) as release height does not allow to accurately predict significant ground concentrations at this distance, especially in stable situations (which is the case in almost all observed peak occurrences, as illustrated in Fig. 9). This is especially true for Doury standard deviations, which simulate a very narrow plume on the vertical, resulting in an underestimation of ground concentrations in 365 the case of elevated release. This was specifically shown in the case of La Hague RP (Connan et al., 2014; Korsakissok et al., 2016). At LRC station (Fig. 8), located farther from the source, this underestimation is much less visible and peaks are better reproduced. There are still, however, peaks that are missed by the deterministic simulations and forecast by some members of the ensemble. This can be explained by the meteorological uncertainties, as detailed in the following section.



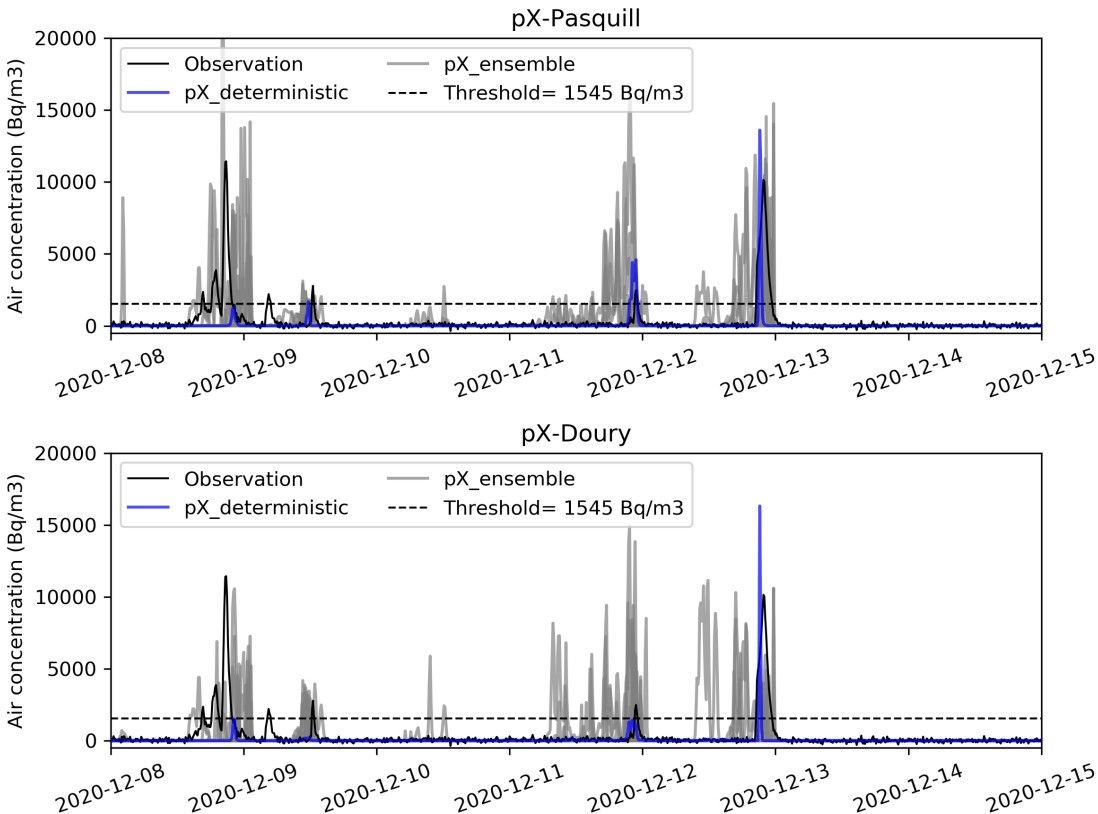

**Figure 8.** The same as in Figure 7, but in the LRC station.

### 4.2.2 Sensitivity to wind and atmospheric stability

In this section, the sensitivity to meteorological variables such as wind and stability is detailed. The aim is to illustrate how small variations in these parameters affect the outputs of atmospheric dispersion simulations. For this purpose, three members of the pX-Pasquill ensemble which have different behaviors are shown. The study is carried out at LRC station, where both air concentration and wind measurements are available. This station is representative of the model behaviour at medium distance, where release conditions are of relatively less importance than meteorological uncertainties. Table 4 summarizes the five

observed peaks from 08 Dec. 2020 to 12 Dec. 2020, when the ensemble behaviour is studied.

   Although the wind forecasts used to generate the three pX simulations in Fig. 9 are sufficiently close to the observation at LRC station around the time of peaks occurrence (as shown in Fig. 10), some events are reproduced either with small errors in timing (i.e. delay/advance of two to three hours) or with errors in intensity (i.e. underestimation/overestimation). This can be a result of local effects on the dispersion simulations. In other words, when one is interested in calculating the activity





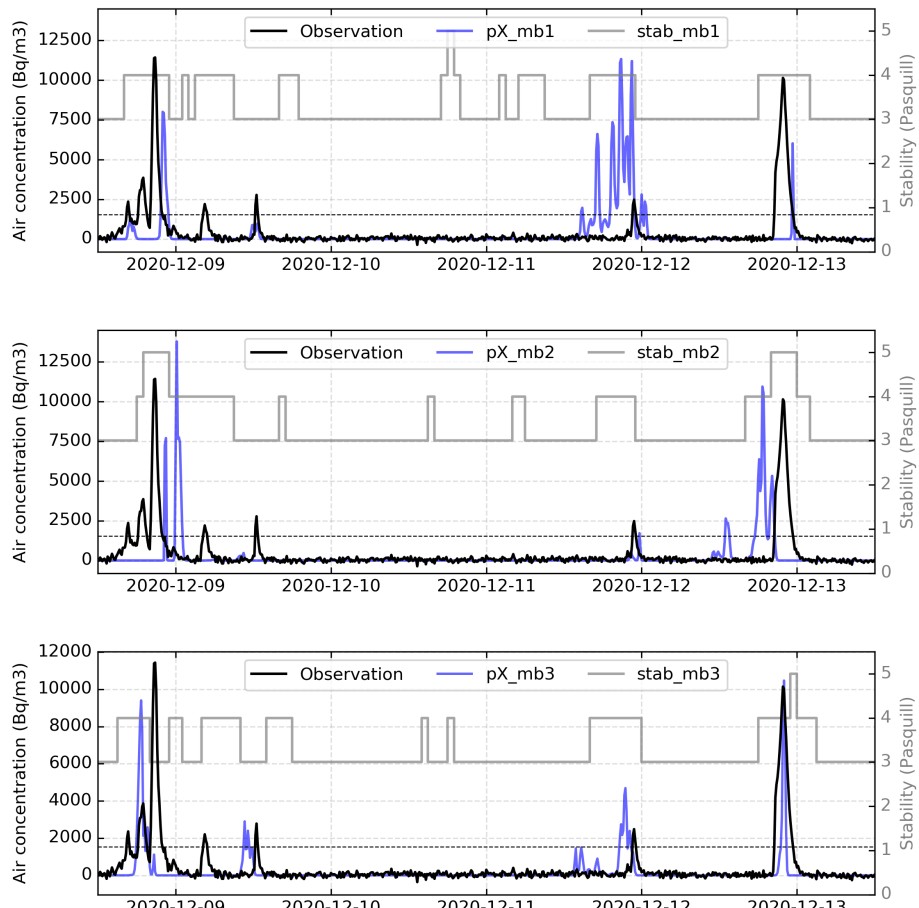

**Figure 9.** The three selected pX simulations of $^{85}$Kr activities, at LRC site, from 08 Dec. 2020 at 12:00 to 13 Dec. 2020 at 12:00. The gray curve represents the hourly Pasquill stability classes calculated from gradient temperature diagnostic. The horizontal dashed line shows the air concentration threshold (1545 $Bq.m^{-3}$) above which peaks are considered.

concentration at a point in space, a small error in the wind speed and/or direction can have a significant impact on the estimation of the peaks timing and intensity, due to the sharp concentration gradient.

Figure 9 also shows the effect of the diagnosed stability classes on the dispersion simulations. Almost all the simulated peaks are associated with stable conditions of the atmosphere ($4^{th}$ or $5^{th}$ Pasquill stability class, corresponding to classes E and F respectively in Appendix A). This may explain the failure of members 1 and 2 to reproduce the third observed peak which is

associated with neutral stability conditions ($3^{rd}$ stability class, corresponding to class D of Pasquill).

In order to better understand the effect of the stability conditions on the pX simulations, a test was carried out by using stationary stability classes in the simulations, using each time one of the three classes obtained by the temperature gradient





|  | Peak 1 | Peak 2 | Peak 3 | Peak 4 | Peak 5 |
|---|---|---|---|---|---|
| Date (UTC) | 08 Dec. 2020, 20:45 | 09 Dec. 2020, 04:30 | 09 Dec. 2020, 12:30 | 11 Dec. 2020, 22:50 | 12 Dec. 2020, 21:50 |
| Observed activities | 11432 | 2204 | 2780 | 2482 | 10145 |
| Member 1 [timing error] | 8000 [+1h15min] | – | – | 11335 [-2h00min] | 6000 [+1h30min] |
| Member 2 [timing error] | 13795 [+3h20min] | – | – | 1714 [+0h50min] | 10945 [-3h10min] |
| Member 3 [timing error] | 9400 [-2h05min] | – | 2900 [-1h50min] | 4690 [-1h20min] | 10468 [+0h10min] |

**Table 4.** The five observed peaks of $^{85}$Kr at LRC station (Figure 9) from 08 Dec. 2020 to 12 Dec. 2020, and the simulated peaks in the three selected members (in $Bq.m^{-3}$), for pX-Pasquill configuration.

diagnostic in the period shown in Fig. 9. The result, shown in Fig. 11, confirms that the diagnosed stability classes has a significant effect, mainly on the simulated intensity. It allows to explain some model failures, such as the peak 3 for the

member 1, but not others. In most cases the $4^{th}$ stability class gives the highest $^{85}$Kr concentration.

In summary, while some detections / non-detections can be easily explained by examining wind speed and direction time series, other features are less predictable on this sole basis, due to the interaction between the variables and the possible accumulation of small direction errors over the plume trajectory.

### 4.3 Statistical evaluation of the dispersion ensemble

#### 4.3.1 Evaluation procedures

It is often a desirable feature for a dispersion model to be able to correctly predict a threshold exceedance. It is particularly useful for decision-making purposes, when protective actions for the population are based on the prediction of zones where a given dose threshold could be exceeded. Evaluating the model performance for this kind of purpose is often based on contingency tables (Daniel and Wilks, 2006) allowing to compare the series of observations and simulations by counting four

features: (i) true positive (TP) when a peak is observed and well simulated, (ii) false negative (FN) when a peak is observed but not simulated, (iii) false positive (FP) when there is no observed but simulated peak and (iv) true negative (TN) when there is no observed and no simulated peak.

The method used by Quérel et al. (2022) is based on this principle and used to evaluate a series of peaks from deterministic simulation against observations. This method consists in evaluating the success/failure of the model for each observed or

simulated peak, including a defined temporal tolerance. However, in the case of an ensemble, the same procedure cannot apply because the are multiple simulations and so unobserved events cannot be well-defined. In addition, it often occurs that the





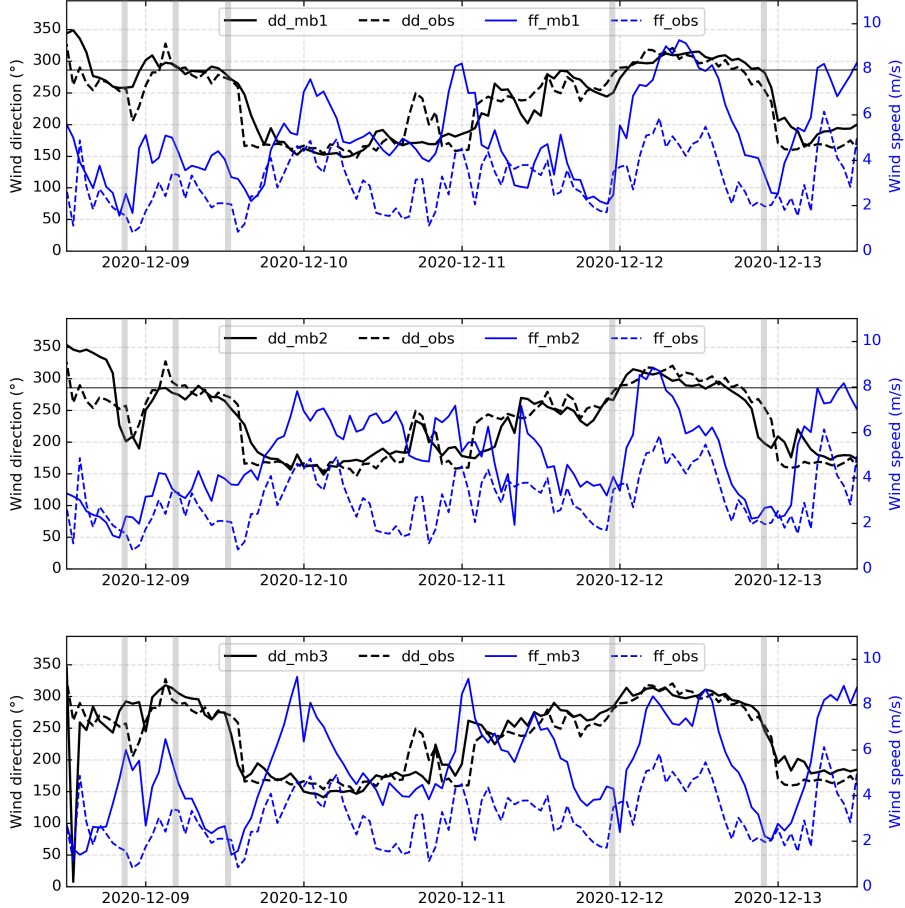

**Figure 10.** The three wind forecasts from AROME-EPS (solid curves) used to generate the pX simulations presented in Figure 9, compared to wind observations stored at LRC (dashed curves). The vertical gray lines shows the occurrence time of the five peaks presented in Table 4. The horizontal line shows the angle of the LRC station with respect to the source (286°), which corresponds to the wind direction that transports the plume from the source to LRC.

FPs from different members constitute a series that exceeds the correlation timescale between peaks (i.e. the temporal offset between two peaks at which they no longer correspond to the same event). Hence, one cannot decide whether all these TPs correspond to the same event or several. To deal with this problem, the method used in this work consists in discretizing the

time series by sliding intervals of length $\Delta t$ and moving time $\tau$ (Fig. 12). For the $k^{th}$ discretization step, the evaluation interval is $[t', t' + \Delta t]$, such that:

$$t' = t_0 + (k-1)\tau \tag{7}$$

Where $t_0$ is the initial time of the time series.



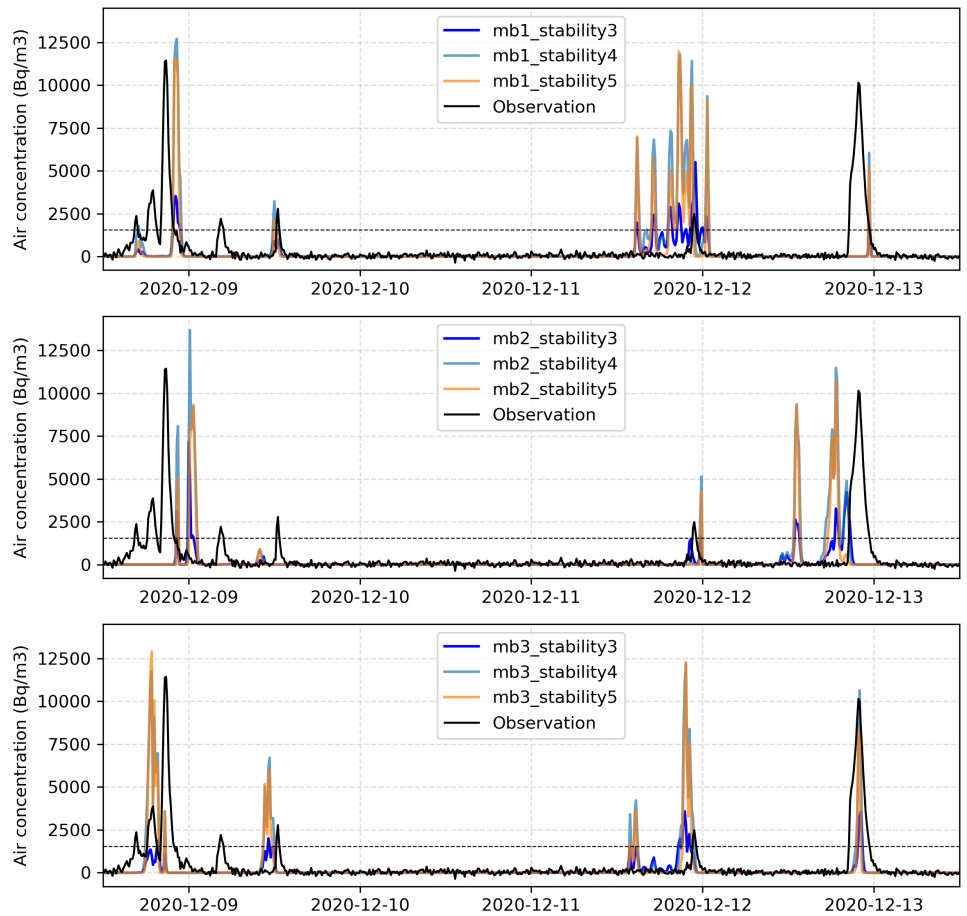

**Figure 11.** The same experiments as in Figure 9, but with three stationary stability classes of Pasquill: $3^{rd}$, $4^{th}$ and $5^{th}$ class.

Then, the maximum values from each ensemble simulation are compared to the maximum observed value in each discretiza-
tion interval. Thus, considering a threshold of 1545 $Bq.m^{-3}$ (corresponding to the detection threshold for air concentration of
$^{85}$Kr) and a given decision threshold $x$ (i.e. the number of members at which the success/failure of the ensemble is considered),
the four features of the contingency tables are defined in this case as follows:

  – TP: when a peak is observed and well simulated by at least $x$ members,

  – FN: when a peak is observed but not simulated by enough members (less than $x$),

– FP: when there is no observed peak but simulated by at least $x$ members,

  – TN: when there is no observed peak but simulated by a number of members less than $x$.




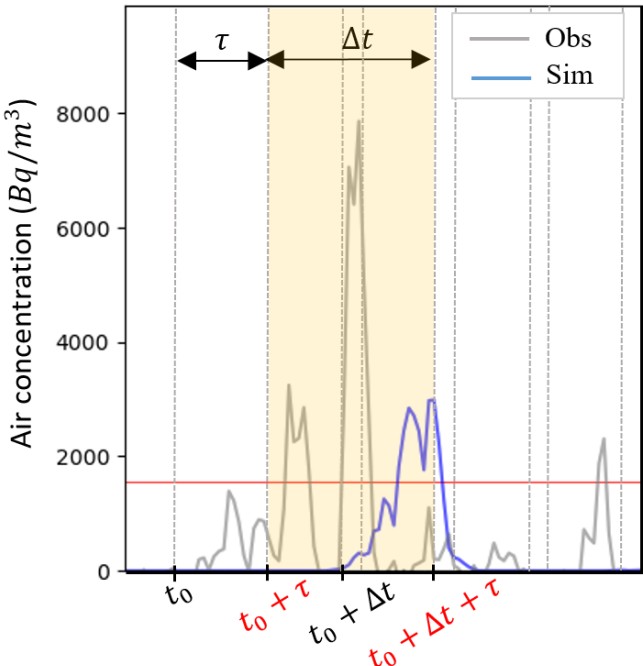

**Figure 12.** Illustration of the temporal discretization method by sliding intervals used in this study. This figure shows the case of the second time interval of length $\Delta t$: $[t_0 + \tau, t_0 + \Delta t + \tau]$.

This method also allows to decrease the number of TN without having a statistically significant impact on the scores thanks to the normalization in the contingency tables. This advantage gives the possibility of using scores that integrate the number of TN, contrary to the classical method which is not adapted to the case of rare events (i.e. the case of non continuous events in 425 time).

Then, the performance of the ensemble is measured using hit rate (H) and false alarm rate (F) metrics (Quérel et al., 2022; Daniel and Wilks, 2006). The hit rate (also called recall) is the fraction of the observed events that are successfully reproduced (Eq. (8)). The false alarm rate is the fraction of the simulated peaks that are not observed (Eq. (9)).

$$H = \frac{TP}{TP + FN} \tag{8}$$

$$F = \frac{FP}{FP + TN} \tag{9}$$

To choose the most representative combination $(\Delta t, \tau)$, we consider that two events are independent if they are more than 3 hours apart. Thus, six combinations $(\Delta t, \tau)$ are tested by the statistical scores below: $(\Delta t = 1\,h, \tau = 1\,h), (\Delta t = 2\,h, \tau = 1\,h), (\Delta t = 2\,h, \tau = 2\,h), (\Delta t = 3\,h, \tau = 1\,h), (\Delta t = 3\,h, \tau = 2\,h)$ and $(\Delta t = 3\,h, \tau = 3\,h)$.

In the case of the AROME-EPS-pX ensemble, there are 16 possible decision thresholds ($x = 1, 2, ..., 16$). In order to identify 435 the most optimal ones, the ROC (Relative Operating Characteristic) curves are commonly used as a graphical summary of





| Aggregated periods | 08 Dec. 2020 to 15 Dec. 2020 | 26 Dec. 2020 to 30 Dec. 2020 | 07 Jan. 2021 to 26 Jan. 2021 | Total |
|---|---|---|---|---|
| Observed peaks | 116 | 92 | 200 | 408 peaks/30days |

**Table 5.** Aggregated time periods for calculating the probabilistic scores of evaluation of pX ensemble and deterministic simulations.

the decision-making skill of an ensemble, by connecting all points $[F(x), H(x)]$ for each decision threshold $x$ (Swets, 1973; Daniel and Wilks, 2006; Raynaud and Bouttier, 2016). In addition, to better capture the internal variation of the performance of the model according to the decision thresholds, the Peirce skill score (PSS) (Peirce, 1884; Daniel and Wilks, 2006) was calculated for each $x$, as follows:

$$PSS(x) = H(x) - F(x) = \frac{TP \times TN - FP \times FN}{(TP + FN) \times (FP + TN)} \tag{10}$$

Note that the $PSS(x)$ corresponds to the vertical distance between the point $[F(x), H(x)]$ of the ROC curve and the no-skill line (i.e. the bisector line, $H = F$). That means that the threshold that presents a better compromise between the probability of detection and the probability of false detection of events corresponds to the one that maximizes the PSS (the closest point to $[F = 0, H = 1]$ in the ROC) (Manzato, 2005, 2007).

Finally, the verification process was performed by aggregating the measurements and simulations of all stations in three periods where a high density of events was recorded (Table 5). This gives a total number of observed threshold exceedance events of 408 over 30 days which is sufficient for the metrics to be statistically robust.

### 4.3.2 Statistical results

Simulations and observations at all stations were aggregated in order to investigate the probabilistic performance of the ensembles, using ROC curves and PSS. Figure 13 shows the results for three combinations $(\Delta t, \tau)$ (the three other cases that were not shown are similar to $(\Delta t = 1\ h, \tau = 1\ h)$ and $(\Delta t = 2\ h, \tau = 1\ h)$). For the deterministic simulations, all the discretization configurations give almost the same false alarm rate (around 6 %) but with large difference for hit rates, with a difference of about 20 % between the best and the worst configuration. The best scores were obtained with the discretization parameters $(\Delta t = 3\ h, \tau = 2\ h)$ for both pX-Doury and pX-Pasquill simulations. This configuration also gave the closest results to the scores obtained with the method of Quérel et al. (2022) for the deterministic simulation (not shown here).

The pX-Pasquill ensembles perform better than pX-Doury, with $PSS_{max} = 0.72$ corresponding to an optimal decision thresholds of 3 and 4 members (against $PSS_{max} = 0.63$ and optimal decision thresholds of 3 members for pX-Doury ensembles). This difference in performance seems normal given that the variation in atmospheric stability conditions is better captured with the Pasquill's stability classes (six classes for Pasquill against two classes for Doury). Besides, the Doury standard deviations were reported to give better results at medium-long distance than at short distance (Korsakissok and Mallet, 2009).

In both stability configurations, the ensemble performs better than the deterministic simulation in a range of seven decision thresholds, which represents almost 50 % of the possible values of the decision thresholds. In addition, the ensemble simula-





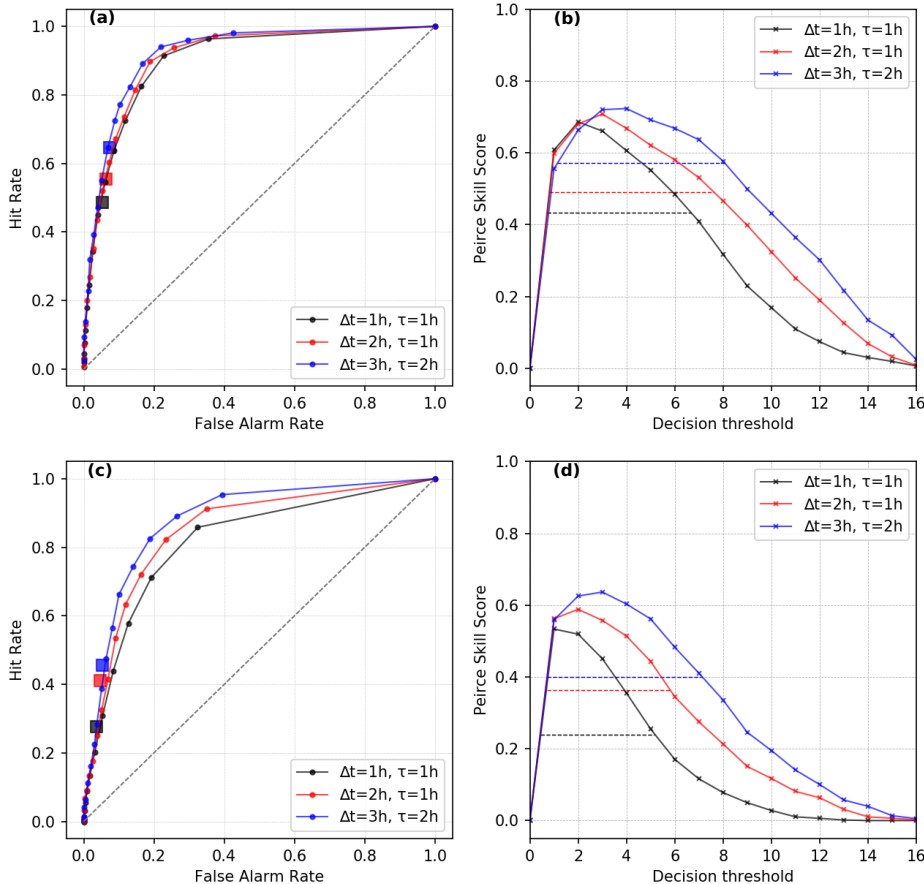

**Figure 13.** ROC curves (a, c) and the PSS as a function of decision thresholds (b, d) of the pX ensemble simulations performed with Pasquill stability classes (a, b) and Doury classes (c, d), by aggregating simulations and observations at all stations. There is one curve for each $(\Delta t, \tau)$: $(\Delta t = 1\ h, \tau = 1\ h)$, $(\Delta t = 2\ h, \tau = 1\ h)$, $(\Delta t = 3\ h, \tau = 2\ h)$. The values of the scores for the deterministic pX simulation are indicated by squares in the ROC curves and by horizontal dashed lines in the PSS curves. The diagonal dashed lines are the no-skill lines $(H = F)$.

tions allows to optimize the decision threshold (Richardson, 2001). These results highlight the robustness of the probabilistic

simulations compared to the deterministic simulation in the process of the prediction of threshold exceedances.

To go further into the analysis of the probabilistic performance of the ensembles, the effect of the distance from the source is investigated in Fig. 14. The most representative discretization parameters $(\Delta t = 3\ h, \tau = 2\ h)$ were used to generate dispersion



simulations with the two diffusion configurations of Pasquill and Doury, by aggregating data for two groups of stations. The first is $-10km$ group which contains stations at distances less than 10 km: PTILH (2 km), Digulleville (2.6 km), Beaumont
(4.2 km) and Gréville (5.2 km). The second is $+10km$ group which contains stations beyond 10 km: Urville (10.4 km), Ludiver (12.7 km), Octeville (17.7 km) and LRC (18 km).

For both groups of stations, pX-Pasquill simulations again gives better scores than pX-Doury, both for deterministic and ensemble pX outputs.

With the Pasquill Gaussian standard deviations (Appendix B), the model performs better in the near-field stations. In this
case, the ensemble is more efficient than the deterministic simulation in 50 % (8 members) of the decision thresholds, against 37.5 % (6 members) for stations located beyond 10 km. In both cases the optimal threshold is 3 members.

With the diffusion laws of Doury, the best scores are obtained also for the group of nearest stations to the source in the case of deterministic simulation. However, for the ensembles there is no significant dependency of the probabilistic scores with respect to the distance from the source.

Taking into account both meteorological and model uncertainties would imply generating an ensemble by also perturbing model parameters (Pasquill/Doury, source elevation, stability). In this perspective, a 32-member super-ensemble was generated by combining pX-Pasquill and pX-Doury ensembles. The result (not shown here) is very similar to the pX-Pasquill ensemble, without any notable added value.

## 5 Conclusions and perspectives

In this study we explored the potential value of using fine-scale spatial and temporal meteorological ensembles to represent the inherent meteorological uncertainties in ADM outputs. To do so, the high-resolution operational forecasts AROME-EPS of Météo-France have been coupled to the Gaussian puff short-range dispersion model pX of IRSN to generate a 16-member dispersion ensemble, which accounts for meteorological uncertainties. This paper also proposes an original method to evaluate the ability of a dispersion ensemble to forecast threshold exceedances, using probabilistic scores. For this purpose, we used an
original data set of continuous $^{85}$Kr air concentration measurements (DISKRYNOC campaign recently conducted by IRSN), along with a well-known source term (every 10 minutes, provided by Orano La Hague RP) and meteorological data (NWP from Météo-France and continuous observations from Météo-France/IRSN).

As a first step, the assessment of the quality of the AROME-EPS forecasts, in the North-Cotentin peninsula (north-western France) was carried out, using meteorological observations, over the two-month period of interest (Dec. 2020-Jan. 2021). Wind
speed and direction are the most influential variables on the transport of a plume through the atmosphere. For this reason, the meteorological ensembles were evaluated in terms of these two meteorological variables. The results of this evaluation showed that the high horizontal, vertical and temporal resolution ($2.5 \times 2.5$ km, 25 vertical levels in the ABL and hourly forecasts) of the AROME-EPS forecasts allow them to correctly represent the uncertainties within ABL.

Then, an ensemble dispersion modeling chain was implemented using the AROME-EPS forecasts as inputs to the pX model.
At this stage, it was necessary to set up a way to combine several 45 h forecasts from different initialization times, and that could



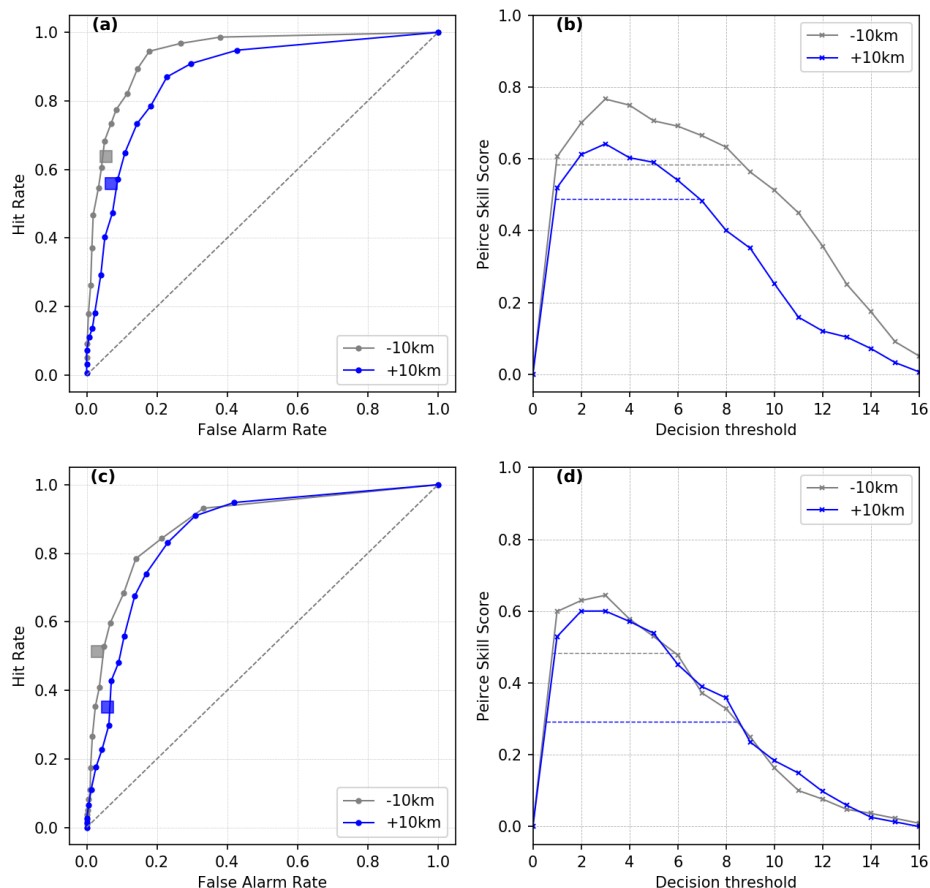

**Figure 14.** ROC curves (a, c) and the PSS as a function of decision thresholds (b, d) of the pX ensemble simulations performed with Pasquill stability classes (a, b) and Doury classes (c, d), by aggregating data in the two groups of stations: $-10km$ (Beaumont, Digulleville, Gréville, and PTILH) and $+10km$ (LRC, Ludiver, Octeville and Urville). There is one curve for each group of stations. The scores are calculated using the most optimal discretization configuration ($\Delta t = 3\,h, \tau = 2\,h$).

be used in the early phase of a nuclear accident. The method proposed in this paper is to use the newest forecast available at the beginning of a day (at 0000 UTC). This approach can be used to span long period, in the case of an emergency, by juxtaposing 24-h successive forecasts. Then, two configurations of dispersion simulations were run, with Pasquill and Doury Gaussian standard deviations. A qualitative assessment of the simulations was first presented, to illustrate the ability of some members of the ensemble to forecast peaks while the deterministic simulations failed. The sensitivity of the results to atmospheric stability




diagnosis was also highlighted. The probabilistic consistency of the two resulting dispersion ensembles was then compared using an innovative method of temporal discretization by sliding intervals, and by calculating two probabilistic scores: ROC curves and PSS. This evaluation process was performed in two parts. First, by comparing the overall performance of the two configurations by aggregating the data from all the measurement stations. In this case the best results were obtained with

Pasquill standard deviations. Secondly, by comparing the performance of the two configurations in the near fields (stations located less than 10 km from the source) and far fields (stations beyond 10 km from the source). The results showed that the Pasquill simulations were still the most consistent with observations. In all cases studied, the best decision threshold is 3 members, and the ensembles performed better than the deterministic simulations. For operational purposes during emergency situations, this result would imply that when 3 or more members of the ensemble forecast a threshold exceedance, protective

actions should be recommended.

One of the limitations of this study is that it evaluates the performance of the dispersion ensembles by only considering its ability to forecast a value above a given threshold, with a temporal tolerance between the simulated and observed peaks. To complete this evaluation, it would be interesting to develop complementary indicators that evaluate the consistency of dispersion ensembles in term of intensity between the simulated and observed peaks. In addition, since $^{85}$Kr is a noble gas,

this work do not investigate the deposition of radionuclides on the ground, a parameter that would be sensitive to uncertainties in rain forecasts. Thus, it will be interesting to apply the approaches developed in this study to the case of another atmospheric tracer that is not an inert gas, such as radon-222 (Quérel et al., 2022).

Another perspective of this study is to work on the clustering of the meteorological ensembles in a perspective of reducing the number of members while keeping the consistency of the dispersion ensembles. This can significantly reduce the computational

time of ADM runs, which is a crucial issue in the case of a real nuclear accident.

## Appendix A:  Atmospheric stability by classes

### Stability classes of Doury (Doury, 1976)

This diagnosis consists in the discretization of the atmospheric stability in two classes: Normal Diffusion (ND) and Low Diffusion (LD). ND corresponds to unstable and neutral situations. LD corresponds to stable situations. In addition, this method

is based only on the vertical temperature gradient, which means that it does not take into account the turbulence of mechanical origin:

$$\frac{\partial T}{\partial z} \begin{cases} \leq -0.5 (°C/100m) & : \text{ND} \\ > -0.5 (°C/100m) & : \text{LD} \end{cases} \tag{A1}$$

### Stability classes of Pasquill (Pasquill, 1961)

The Pasquill classes allows to discretize the atmospheric stability in six classes from A (very unstable, coded by 0) to F (very

stable, coded by 5). In this article we have used two diagnostics of calculating of the Pasquill classes:





| 10-m wind speed (m.s$^{-1}$) | Day Surface solar radiation downwards w/m2 | | | Night Total cloud cover (%) | |
|---|---|---|---|---|---|
| | Strong ]700, +∞] | Moderate [350, 700] | Low [0, 350[ | [4/8, 7/8] | [0, 3/8] |
| <2 | A | A-B | B | F | F |
| 3 | A-B | B | C | E | F |
| 3 - 5 | B | B-C | C | D | E |
| 5 - 6 | C | C-D | D | D | D |
| >6 | C | D | D | D | D |

**Table A1.** Pasquill classes according to method of Turner.

| Atmospheric stability | Stability classes of Pasquill | $\partial T/\partial z$ | Stability classes of Doury |
|---|---|---|---|
| Unstable | A | ]-∞, -1.9[ | Normal Diffusion (ND) |
| | B | [-1.9, -1.7[ | |
| | C | [-1.7, -1.5[ | |
| Neutral | D | [-1.5, -0.5[ | |
| Stable | E | [-0.5, 1.5[ | Low Diffusion (LD) |
| | F | [1.5, +∞[ | |

**Table A2.** Pasquill classes according to method of temperature gradient, and correspondence with the classes of Doury.

- **Method of Turner** (Turner, 1969): Based on the 10-m wind speed and the surface solar radiation downwards (during daytime) or total cloud cover (at night) (Table A1). This diagnostic has the advantage of taking into account the two origins of turbulence: Mechanical (wind) and thermal (solar radiation).

- **Method of temperature gradient** (Seinfeld and Pandis, 1998): Based on the temperature difference over 100m (Table A2). This diagnosis does not take into account the turbulence of mechanical origin, but it captures variations in stability conditions better than Doury diagnosis.

## Appendix B: Formulas for Gaussian standard deviations

The aim is to calculate the evolution of standard deviations of the concentration distribution, which is given by:

$$\sigma(t + dt) = \sigma(t) + \frac{d\sigma}{dt} dt = \sigma(t) + \frac{d\sigma}{dx} U dt \tag{B1}$$



Where $t$ and $x$ are time and distance since the emission of the puff, respectively. $U$ is the speed of advection of the puff.

Thus, The problem is deferred to the determination of $\frac{d\sigma}{dx}$ (or $\frac{d\sigma}{dt}$) in each time step $t$. For a given standard deviation law, we have: $\sigma(x) = f(x, \alpha_1, \alpha_2, ..., \alpha_n)$, where $\alpha_i$ are parameters that depend on the atmospheric stability, and are determined empirically. Then, $\frac{d\sigma}{dx}$ can be expressed as follows:

$$
\begin{cases}
\frac{d\sigma}{dx} = \frac{\partial f}{\partial x}(x, \alpha_1, \alpha_2, ..., \alpha_n) \\
x = f^{-1}(\sigma)
\end{cases}
\Leftrightarrow
\begin{cases}
\frac{d\sigma}{dx} = \frac{\partial f}{\partial x}(f^{-1}(\sigma), \alpha_1, \alpha_2, ..., \alpha_n) \\
\quad = \phi(\sigma, \alpha_1, \alpha_2, ..., \alpha_n)
\end{cases}
\tag{B2}
$$

A similar reasoning is possible for $\frac{d\sigma}{dt}$. Depending on the complexity of the function $f$, it will be more or less easy to express the function $\phi$.

– **Pasquill laws**:

$$
\sigma = ax^b + c \Rightarrow \frac{d\sigma}{dx} = ab\left(\frac{\sigma}{a}\right)^{\frac{b-1}{b}}
\tag{B3}
$$

The parameters $a$, $b$ and $c$ are determined according to the Pasquill stability classes.

– **Doury laws**:

$$
\sigma = At^k t^{k-1} \Rightarrow \frac{d\sigma}{dx} = kA\sigma^{\frac{k-1}{k}}
\tag{B4}
$$

The parameters $A$ and $k$ are determined according to the Doury stability classes, and defined by step of time.

*Code and data availability.* The Arome and Arome-EPS data can be accessed by https://donneespubliques.meteofrance.fr/. IRSN data (radiological and meteorological) are available on demand. To get statistical calculation code and the plotting code please contact the corre-
sponding author.

*Author contributions.* The project was conceptualized and supervised by MP and IK. Formal analysis and development of the calculation codes were carried out by YE. The radiological measurement campaign was conducted by OC. All the authors contributed to the discussion of the results and to writing the article.

*Competing interests.* The authors declare that they have no conflict of interest.

*Acknowledgements.* The authors thank Orano RP for providing source term and environmental measurement data ($^{85}$Kr and meteorological measurements). The authors thank Emmanuel Quentric for his review, and Arnaud Quérel and Pierrick Cébron for discussions and suggestions on the use of statistical indicators.



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
