# Peer review of "Combining short-range dispersion simulations with fine-scale meteorological ensembles: probabilistic indicators and evaluation during a 85Kr field campaign"

_EGUsphere, 2022_

## Author Response (AR1)

**Acknowledgements**

We thank the reviewers very much for their constructive comments which helped to improve the quality of the paper. In this letter below, we answer to all comments and explain how they have been addressed in the revised manuscript. We hope that this new version may be accepted for publication in Atmospheric Chemistry and Physics.

**General Comments reviewer 1**

The authors demonstrate the value of ensemble meteorology by showing how it can be used to model the uncertainty in the dispersion of material released from a known source. Where previous studies have focussed on long-range dispersion and used meteorology from global NWP models this study examines the use of a high-resolution (2.5km) ensemble NWP model covering a limited area to provide meteorological input to a dispersion model. In addition, the study focuses on modelling the dispersion of material that is regularly discharged from a reprocessing plant and compares the model results to observations over a period of two months. This field campaign along with the meteorological model and the dispersion model are clearly described within the paper.

My main complaint about this paper is that, for me, it covers too many topics. This has two impacts, first I am distracted from the main results and second the secondary topics are not covered in great detail so I am left with too many questions, as can be seen by the length of the specific comments section. There are two main sections which take my attention away from the main results. The first is the work looking at different methods to compute stability. The second is the consideration of how to model dispersion over time period which are longer than a single meteorological forecast. Both of these are interesting topics but, I feel they would be better placed in separate papers where they can be discussed more fully. I have included my specific comments on both of these sections in the specific comments.

**General Comments – answer from authors**

In the revised version of the paper, these comments have been addressed in two ways:

1/ less emphasis has been made on the comparison between the Gaussian standard deviation formulas (Pasquill vs. Doury). In order to make the reader less distracted, and given that the main objective of the paper is the evaluation of the ensemble predictions, the section on statistical results (section 4.3.2) has been lightened by focusing only on the Pasquill method.

2/ the explanation of how and why we model dispersion over a time period which are longer than a single meteorological forecast has been improved. We believe it is an important aspect of the simulation set-up, that is addressed in a short section of the manuscript.

Both aspects are covered in detail in the specific comments, and we hoped that this addresses the reviewers' concerns.

**Specific Comments**

• In the Introduction many papers are mentioned and in some cases the work carried out is described. It would also be helpful to understand the results or outcomes of the work in those papers. For example, the authors note that evaluations of dispersion ensembles were performed by Le et al, (2021) and De Meutter and Delcloo (2022) but they don't say whether the ensembles were found to perform well or whether the use

of ensembles provided more information. Similarly the authors mention the works of Galmarini et al,. (2004a and b) in performing multi-model ensembles but do not say anything about the findings of those works.

Changes made in the text, Line 59 :

In Le et al. (2021) an De Meutter and Delcloo (2022), an evaluation of the dispersion ensembles was performed by comparison to radiological observations in the environment, *and the results illustrate the added value of the use of weather ensembles for dispersion simulations*.

Changes made in the text, Line 74 :

This approach was extensively investigated in Galmarini et al. (2004a, b) by using a set of different ADM to construct an ensemble of simulations, either with identical or different input data, to represent the modelling uncertainties, and the results showed that the ensemble simulations allows to reduce the uncertainty related to the deterministic simulation.

• Line 50: I'm not sure "coarse" is the appropriate word to use here as one of the studies referenced in the previous paragraph used meteorological data at a resolution of 2.5km which is not generally considered to be a coarse resolution.

Changes made in the text, Line 51 :

All these studies were carried out at long distance and the ensembles used to represent weather uncertainties had coarse spatial and temporal resolution, except Leadbetter et al. (2022) who used also fine-scale weather ensembles with a horizontal resolution of about 2.5 x 2.5 km and 70 vertical levels.

• Line 82: Is it possible to define "reasonable" in reference to the 85Kr release? Is the error on the release rate known?

The quantity of 85Kr released to the atmosphere is measured by the operator with a temporal accuracy of 10 minutes (confidential data) and an uncertainty of about 10% on the activity measured (calculated by the difference between the data obtained on 2 measurement channel during the period of release).

Changes made in the text, Line 86 :

*The main sources of the* 85*Kr in the atmosphere are reprocessing plants of spent nuclear fuel, from which the* 85*Kr release can be known with accuracy (described in section* 2.2).

Paragraph added, Lines from 148 to 151 :

The activity in 85Kr released from the factory by the stacks (confidential data) is known with a time step of 10 minutes and an uncertainty of measurement of the order of 10% in period of release (two channels of measurements for each stack). The discharge being intermittent, this 10-min time step ensures a precision indispensable for atmospheric dispersion studies. From 2019 to 2021, annual releases of the 85Kr varied from 294 to 379 PBq/year (Orano, 2021).

• Line 115: Is it possible to provide an approximate activity concentration for the amount of 85Kr naturally present in the environment or a ratio of the 85Kr present in the environment to the amount of 85Kr released by the reprocessing?

**Changes made in the text Lines from 120 to 124:**

Background levels of 85Kr in the atmosphere, excluding an industrial plume, are currently below 2 Bq.m-3 (Bollhofer et al, 2019). In nearby fields in the plume around the RP of La Hague (about 0-2 km), activities can reach 100,000 Bq/m3 (Connan et al 2014). At distances of the order of 20 km, the maximum measurable activities are generally less than 10000 Bq/m3 and beyond a few tens of km of RP, the activities in 85Kr are too low to be measurable in real time (Connan et al 2013).

• Line 117: Similar to line 82; is it possible to define "reasonable" in reference to the 85Kr release? Is the error on the release rate known?

See comments above

• Line 125: In describing the terrain around La Hague as complex is it possible to provide values for the maximum and minimum elevations to provide meteorological readers with a reference point for how the terrain might affect the wind speed and direction?

Changes made in the text, Line 133 :

*The North-Cotentin peninsula of La Hague is a rocky area of approximately 15 km located at 190 m a.s.l above cliffs, surrounded by the sea less than 5 km in most directions (Fig. 1).*

• Line 136 and 137: For me the availability of data at a 10-minute resolution doesn't, on its own, constitute an accurate and reliable source term. I would be interested to know the uncertainty on the measurements relative to the amount of material released.

The following sentence is deleted: Line 136 and 137 [*The sum of the amounts of 85Kr released from UP2 and UP3 units, over regular 10 minutes.... and reliable source term.*] Measurement uncertainty relative to source term was added in Line 149 (see comment above).

• Table 2: Would it possible to add the temporal resolution of the met data to the table? I think this is mentioned later on in the text but it would be helpful to include it in this table too.

Changes made in the table 2.

• Section 3.1: There are a large number of different skill scores which could be used for the verification of both deterministic and ensemble predictions. Would it be possible for the authors to include an explanation of why bias and spread-skill were chosen?

The choice is based on subsequent work by the meteorological community. The most commonly used scores for evaluating the reliability of ensembles (the ability of the meteorological ensembles to represent realistic uncertainties) are the spread-skill ratio and rank diagrams (not shown in the paper), which are complementary scores. On the other hand, the bias allows to identify the systematic errors of the weather predictions, as explained in the text.

**Changes made in the text, Line 267:**

For this purpose, two common scores, among others, used by the meteorological community for the evaluation of ensemble reliability, have been calculated based on the observations of 3D-wind speed and direction...

• Figure 3: In the text the authors mention that there is a diurnal cycle in the bias, but I find this difficult to see because the bias shares the same axis as the mean values. Would it be possible to place the bias on a separate axis to the mean values?

Changes made in the Figure 3.

• Section 4.1.1: I am very surprised that it is necessary to use more than one 24-hour forecast for this study. The furthest observation point is situated